# Gene Electrotransfer into Mammalian Cells Using Commercial Cell Culture Inserts with Porous Substrate

**DOI:** 10.3390/pharmaceutics14091959

**Published:** 2022-09-16

**Authors:** Tina Vindiš, Anja Blažič, Diaa Khayyat, Tjaša Potočnik, Shaurya Sachdev, Lea Rems

**Affiliations:** 1Faculty of Electrical Engineering, University of Ljubljana, Tržaška 25, 1000 Ljubljana, Slovenia; 2Institute for Multiphase Processes, Leibniz University Hannover, An der Universität 1, 30823 Garbsen, Germany; 3Lower Saxony Centre for Biomedical Engineering, Implant Research and Development, Stadtfelddamm 34, 30625 Hannover, Germany

**Keywords:** localized electroporation, gene electrotransfer, plasmid, transfection, cell culture insert, numerical modeling, Chinese hamster ovary cells, myoblasts, C2C12 cell line, H9C2 cell line

## Abstract

Gene electrotransfer is one of the main non-viral methods for intracellular delivery of plasmid DNA, wherein pulsed electric fields are used to transiently permeabilize the cell membrane, allowing enhanced transmembrane transport. By localizing the electric field over small portions of the cell membrane using nanostructured substrates, it is possible to increase considerably the gene electrotransfer efficiency while preserving cell viability. In this study, we expand the frontier of localized electroporation by designing an electrotransfer approach based on commercially available cell culture inserts with polyethylene-terephthalate (PET) porous substrate. We first use multiscale numerical modeling to determine the pulse parameters, substrate pore size, and other factors that are expected to result in successful gene electrotransfer. Based on the numerical results, we design a simple device combining an insert with substrate containing pores with 0.4 µm or 1.0 µm diameter, a multiwell plate, and a pair of wire electrodes. We test the device in three mammalian cell lines and obtain transfection efficiencies similar to those achieved with conventional bulk electroporation, but at better cell viability and with low-voltage pulses that do not require the use of expensive electroporators. Our combined theoretical and experimental analysis calls for further systematic studies that will investigate the influence of substrate pore size and porosity on gene electrotransfer efficiency and cell viability.

## 1. Introduction

Delivery of nucleic acids into living cells is one of the key steps in genetic engineering, be it for biological studies based on live-cell imaging, cellular manipulation, genome editing, or various medical applications [1]. This delivery is often achieved with the use of viral vectors, which are highly efficient; however, pre-existing immunity and immune reactions following treatment with viral vectors offer limited safety [2]. Electroporation is one of the main non-viral methods for intracellular delivery of nucleic acids, wherein pulsed electric fields are used to transiently permeabilize the cell membrane, allowing enhanced transmembrane transport [3,4]. The use of electroporation for intracellular delivery of nucleic acids is often termed gene electrotransfer. Both DNA and RNA molecules can be delivered into cells with gene electrotransfer, but plasmid DNA has been most commonly used [5].

Conventional gene electrotransfer in vitro is performed as bulk electroporation by placing a cell suspension or cell monolayer with added plasmid between two electrodes and applying short high-voltage electric pulses. A disadvantage of this method is heterogenous permeabilization of cells, since bulk electroporation is influenced by cell properties, including cell size, shape, and membrane lipid composition, which change along the cell cycle [6,7,8]. Therefore, even for an optimal electric field strength leading to efficient gene electrotransfer, a considerable fraction of cells unavoidably becomes extensively electroporated, leading to cell damage and cell death [9]. Furthermore, delivery of high-voltage electric pulses during bulk electroporation can cause electrochemical reactions at the electrode–electrolyte interface, such as reactive oxygen species formation and metal ions release from the electrodes, which can all lead to decreased cell survival and gene electrotransfer efficiency [10].

To minimize cell damage and maximize the control of the gene delivery process, several concepts have been proposed, first based on microstructured [11,12,13] and later on nanostructured [14,15,16,17,18,19,20,21,22,23] geometries, which localize the electric field over a small region of the cell membrane. One of the first nanostructured concepts was based on a series of two microchannels connected by a nanochannel [14]. A cell within one of the microchannels was positioned close to the nanochannel, e.g., with optical tweezers, whereas solution containing plasmid DNA (or other biomolecules) was placed in the opposite microchannel. Since the nanochannel presented the greatest electrical resistance within the system, most of the voltage applied during an electric pulse dropped over the nanochannel, resulting in a greatly enhanced electric field within the nanochannel. This enhanced electric field enabled electrophoretic injection of charged molecules from one microchannel, across the nanochannel, into the cell within the other microchannel. By changing the pulse duration, this approach enabled control over the number of biomolecules delivered into the cell while preserving cell viability. Subsequently, various other concepts were proposed, including cells growing on substrates with hollow nanostraws [15,20] and cells growing on nanofabricated porous substrates [18]. The disadvantage of these approaches, however, is that they require expertise on micro/nanofabrication and access to clean room facilities, which are often not available to cell biology labs.

Recently, Mukherjee et al. [21] and Cao et al. [22] proposed systems based on a commercially available track-etched polycarbonate substrate (otherwise used as water-filter membrane) containing pores with a diameter of 200 nm and 100 nm, respectively. These porous substrates were embedded into a custom polydimethylsiloxane (PDMS) holder and coated with poly-L-lysine or fibronectin to improve cell adherence. The cells were then either grown or sedimented on top of the substrate. Application of electric pulses across the porous substrate enabled delivery of nucleic acids, functional proteins, and Cas9 single-guide RNA ribonucleoproteins into both adherent and suspension-based cells [22]. For plasmid DNA, ~50–80% transfection efficiency and >95% cell viability was achieved [22]. Their studies inspired us to test whether similar success could be achieved with tracked-etched porous membranes embedded within commercial cell culture inserts that are primarily used for drug transport, cell invasion, chemotaxis, and motility studies. The first foreseen advantage of such inserts for gene electrotransfer is that they are already sterilized and precoated for optimal cell attachment. Second, they are pre-attached to a holder, and consequently, require no additional (micro)fabrication procedures to assemble the insert for electroporation. Third, they can be combined with inverted optical microscope when choosing transparent polyethylene terephthalate (PET) membranes. A possible problem of such inserts is that their minimum pore diameter is 0.4 µm, which is 2–4× larger compared to pore diameters used in previous nanoporous systems [14,21,22].

Therefore, in our study, we first used multiscale numerical modeling to determine the membrane pore size, electric pulse parameters, and electrode configuration that are expected to result in localized electroporation and successful gene electrotransfer. Based on the numerical results, we designed a simple system consisting of an insert, a multiwell plate, and a pair of platinum/iridium wires as electrodes. We then experimentally tested this device, assessing the efficiency of plasmid electrotransfer and cell survival in different cell lines, including Chinese hamster ovary (CHO) cells, rat cardiac myoblasts H9C2, and mouse myoblasts C2C12. We obtained up to 44% average transfection efficiency, which is somewhat lower compared to Cao et al. [22], but comparable to that achievable with conventional bulk electroporation. We discuss advantages and limitations of our system, and how gene electrotransfer systems based on porous substrates should be further studied to better understand and optimize the gene delivery process.

To avoid confusion and for a clear distinction in terminologies, throughout the manuscript, we use the term ‘substrate pores’ for pores in the insert’s porous substrate and ‘electropores’ for pores/defects created in the cell membrane due to electroporation.

## 2. Materials and Methods

### 2.1. Numerical Modeling

Numerical modeling was carried out on two different geometric scales. We first modeled electroporation of a single cell on a small fraction of the porous substrate to investigate how the size of substrate pores and the voltage across the substrate influence cell electroporation. However, the voltage across the substrate is not equal to the voltage applied to the electrodes, since part of the applied voltage drops over the conductive aqueous solutions. Therefore, we also built a model of the full insert system, which allowed us to determine how the voltage across the substrate depends on the electrode geometry and the applied voltage. All numerical simulations were carried out in Comsol Multiphysics 6.0 based on the finite element method.

#### 2.1.1. Model of a Cell on Top of the Porous Substrate

The model consists of a cell positioned on top of the porous substrate and surrounded by the extracellular liquid, as shown in Figure 1a. The model can be viewed as a “unit cell” that is periodically repeated along the entire substrate. Two cell sizes were considered, which roughly cover the size range of typical adherent mammalian cell cultures, although the cell size/geometry does not have a large influence on the modeling results for our systems, as shown later in Section 3.1. The pores within the substrate are arranged in an array that mimics the average porosity of the substrates used in experiments. There is a gap of 100 nm between the substrate and the cell membrane. This gap distance was chosen based on previous studies, showing that the average gap between a fibronectin-coated surface and monocyte membrane and is ~80 nm [24], whereas for HEK and HL-1 cells, this distance is ~100 ± 50 nm [25]. We confirmed that the numerical results are not very sensitive to the gap distance, when varied between 50 and 150 nm (Appendix A).

The top and bottom side of the model were assigned an electric potential that results in a 10 ms-long voltage pulse. Since the greatest electrically resistive element in this model is the substrate, practically the entire assigned electric potential difference (i.e., voltage) drops across the substrate (and partially the adjacent cell membrane, where present). The electric potential distribution *V* in the intracellular and extracellular liquid is calculated by:(1)$$\nabla \cdot \left[\left(\sigma_{i,e}+\varepsilon_{i,e}\frac{\partial }{\partial t}\right)\nabla V_{i,e}\right]=0$$
where *σ_i,e_* and *ε_i,e_* denote, respectively, the conductivity and the dielectric permittivity of the intracellular (subscript *i*) or extracellular (subscript *e*) liquid. The PET material of the substrate was modeled as an ideal insulator. The cell membrane was modeled via a boundary condition, which describes the continuity of the normal component of the electric current density **J_e_** across the membrane:(2)$$n\cdot J_{e}=\left(G_{cm}+G_{ep}\right)U_{cm}+C_{m}\frac{\partial U_{cm}}{\partial t}$$
where **n** denotes the unit vector normal to the membrane surface, transmembrane voltage *U_cm_* is the difference between the electric potentials on the two sides of the cell membrane, *U_cm_* = *V_i_* − *V_e_*, *G_cm_* and *C_cm_* denote the passive membrane conductance and membrane capacitance, respectively, and *G_ep_* denotes the increase in membrane conductance due to electroporation, which is modeled by the following differential equation [26]:(3)$$\frac{dN}{dt}=\alpha e^{{\left(U_{cm}/V_{ep}\right)}^{2}}\left(1-\frac{N}{N_{0}}e^{-q{\left(U_{cm}/V_{ep}\right)}^{2}}\right)$$
where *N* denotes the density of electropores formed in the cell membrane by the electric field, *N*_0_ is the electropore density at *U_cm_* = 0 V, and parameters *α*, *q*, and *V_ep_* describe the characteristics of the electroporation process. The increase in the cell membrane conductivity due to electroporation is calculated as [27]:(4)$$G_{ep}=N\frac{2\sigma_{p}\pi r_{p}{}^{2}}{\pi r_{p}+2d_{m}}$$
where *r_p_* and *σ_p_* are the radius and conductivity, respectively, of a single electropore. The fully coupled system of Equations (1)–(4) was solved in the time domain using the default Comsol solver with the “strict” method of time stepping, following previous work [28]. The values of the model parameters are shown in Table 1. Although our model neglects the expansion of electropore size that can occur due to electric field, as considered in a previous study [21], it is able to spatially and temporally resolve electroporation over the entire cell membrane.

#### 2.1.2. Model of the Full Insert System

The geometry of the numerical model of the full insert system was designed to resemble the desired experimental configuration. There are many different commercially available cell culture inserts. We decided to use inserts with transparent polyethylene terephthalate (PET) substrates, since they enable visualization of the cells with an inverted optical microscope both in transmission and fluorescence mode. We further decided to experiment with the smallest commercially available inserts, i.e., inserts for 24-well plates, since we aimed to perform many tests and we preferred the use of small samples. For electrodes we used platinum/iridium (90/10) wires with diameter of 0.5 mm. These wires can be folded in arbitrary shapes, while being rigid enough to maintain their shape after folding. We initially aimed to use titanium wire electrodes (99.7% pure), but we encountered problems due to their electrochemical oxidation. In some of the previous studies investigating gene electrotransfer using porous substrates, the substrates were positioned either on indium tin oxide (ITO) [20] or titanium plate electrode [22]. However, in part of our experiments, we aimed to visualize the cells on the microscope during the pulse delivery, for which the chosen inserts containing the top electrode were not sufficiently stable when positioned on the microscope stage on top of a glass slide. Therefore, we decided to use wire electrodes and fix them to a companion 24-well plate that is specialized for the chosen inserts. Our numerical calculations demonstrated that, as long as we use a spiral bottom electrode, we can achieve a homogenous voltage along the surface of the insert substrate, while still being able to visualize the cells during electroporation with an inverted microscope.

The geometry of the numerical model of the full insert system is shown in Figure 1b. The insert is modeled as a cylinder containing the top liquid (e.g., growth medium). The porous substrate is located at the very bottom of the insert and its surface area matches the area available for cell growth. The insert is positioned in a well of a 24-well plate, which is filled with the bottom liquid (e.g., plasmid solution). There is a distance of 0.8 mm between the bottom of the well and the substrate, which is defined by the geometry of the companion 24-well plate (the inserts have two flanges with which they hang on the top edge of the well). The insert contains the top wire electrode and the well contains the bottom wire electrode. The minimum distance between the top electrode and the porous substrate is 2 mm. The geometry of the electrodes shown in Figure 1b resembles the final configuration; however, we numerically tested different electrode geometries.

The electric potential distribution in the top and bottom liquid was determined by computing the steady state solution to Equation (1). The substrate was modeled via a boundary condition, which is similar to Equation (2) and is given in Equation (5):(5)$$n\cdot J_{e}=G_{subs,eff}U_{subs}$$
where *U_subs_* is the voltage across the substrate and *G_subs,eff_* is the effective conductance of the substrate:(6)$$G_{subs,eff}=\rho_{subs}\frac{2\sigma_{e}\pi {\left(r_{p,subs}\right)}^{2}}{\pi r_{p,subs}+2d_{subs}}$$
where *ρ_subs_* is the porosity (number of pores per unit area) and *r_p,subs_* is the radius of a substrate pore.

### 2.2. Experiments

#### 2.2.1. Cell Culture

For the experiments, we used Chinese hamster ovary cells CHO-K1, rat cardiac myoblasts H9C2, and mouse myoblasts C2C12, all from the European Collection of Authenticated Cell Cultures (cat no. #85051005, #88092904, and #91031101, respectively). The passage numbers used for experiments were 10–28, 12–26, and 7–17 for CHO, H9C2, and C2C12, respectively. All cell lines were grown in 25 or 75 cm^2^ culture flasks in appropriate growth medium in a humidified atmosphere at 37 °C and 5% CO_2_ (CHO) or 10% CO_2_ (H9C2 and C2C12), and were routinely passaged every 2 to 4 days. To plate the cells into cell culture inserts, the cells were first trypsinized and counted. Afterwards, 0.8 × 10^4^ (CHO) or 1.0 × 10^4^ (C2C12 and H9C2) cells in 300 μL of appropriate cell growth medium were added to each insert, whereby the insert was prepositioned into a 24-well plate (TPP #92424, Trasadingen, Switzerland) containing 700 μL of cell growth medium. We used inserts with PET substrates containing pores with either 0.4 µm or 1.0 µm diameter (Falcon, #353095 or #353104, Conrning Inc., Corning, NY, USA). The cells were further grown for 2 days prior to gene electrotransfer.

#### 2.2.2. Plasmid

A 4.7 kb plasmid pEGFP-N1 (Clontech Laboratories Inc., Mountain View, CA, USA) encoding enhanced green fluorescent protein (eGFP) under the control of CMV promotor was used. Plasmid DNA was amplified using Escherichia coli and isolated with HiSpeed Plasmid Maxi Kit (Qiagen, Hilden, Germany). Plasmid concentration in stock solution was spectrophotometrically determined at 260 nm and ranged between 5−9 mg/mL, depending on the batch. The stock solution was stored at −20 °C.

#### 2.2.3. Aqueous Solutions

CHO cells were grown in HAM’s F-12 Nutrient Mix (Sigma-Aldrich #N6658, Merck Group, Darmstadt, Germany), whereas H9C2 and C2C12 were grown in Dulbecco’s Modified Eagle Medium (Sigma-Aldrich #D6546). All growth media were supplemented with 10% fetal bovine serum (Sigma-Aldrich #F9665 for CHO and C2C12, Sigma-Aldrich #F2442 for H9C2), L-glutamine (Sigma-Aldrich #G7513; 1 mM for CHO, 4 mM for H9C2 and 2 mM for C2C12), and antibiotics penicillin-streptomycin (0.01%, Sigma-Aldrich #P0781) and gentamycin (0.1%, Sigma-Aldrich #G1397). Trypsin-EDTA solution (Sigma-Aldrich #T3924) was used for detaching cells from surface. Before trypsinization, cells were washed with physiological saline (0.9% NaCl, B. Braun, Melsungen, Germany). The propidium iodide (PI) staining solution was prepared by dissolving PI stock solution (1 mg/mL in deionized water, Life Technologies #P1304MP, Carlsbad, CA, USA) in Live Cell Imaging Solution (LCIS) in ratio 1:10. LCIS (Molecular Probes, Life Technologies #A14291DJ) is an optically clear physiological solution mimicking the ionic composition of growth media: 140 mM NaCl, 2.5 mM KCl, 1.8 mM CaCl_2_, 1.0 mM MgCl_2_, and 20 mM HEPES at pH 7.4. Plasmid solutions with target concentrations for gene electrotransfer were prepared by diluting the plasmid stock solution (see Section 2.2.2) to 100 µg/mL or 500 µg/mL. For CHO cells, the plasmid was dissolved in LCIS. For C2C12 and H9C2 the plasmid was dissolved in phosphorate-buffered saline (PBS) consisting of 137 mM NaCl (Sigma-Aldrich #S9888), 2.7 mM KCl (Sigma-Aldrich #P3911), 10 mM Na_2_HPO_4_ (Sigma-Aldrich #S9763), and 1.8 mM KH_2_PO_4_ (Sigma-Aldrich #P0662) at pH 7.4, since preliminary results showed better transfection efficiency for H9C2 in PBS compared to LCIS (Appendix A).

#### 2.2.4. Electric Pulses

The ELECTRO cell B10 electroporator (Leroy Biotech, Saint-Orens-de-Gameville, France) was used to supply electrical pulses. The voltage and current applied to the sample were measured and monitored on LeCroy Wavepro 7300A (3 GHz) oscilloscope with a PP007-WS 10x probe (LeCroy, DC-500 MHz) or ADP305 High Voltage Differential Probe (LeCroy, DC-100 MHz) and AP015 Current Probe (LeCroy, DC-50 MHz) during the experiment. We delivered 1–12 pulses with a length of 10 ms at different voltages in the range between 10 V and 40 V and at a pulse repetition frequency of 1 Hz. The shape of the delivered pulses was not fully rectangular, since the generator’s capacitors began to discharge during the pulse (Appendix A). Throughout the paper, the peak applied voltage is reported as the pulse amplitude. For pulse delivery, the inserts were always positioned on flanges of the companion 24-well plate (Falcon #353504) to maintain a controlled distance of 0.8 mm between the bottom of the well and the substrate (see Section 2.1.2).

#### 2.2.5. Propidium Uptake

To characterize electroporation of cells growing on inserts, we first monitored the kinetics of propidium iodide (PI) uptake during pulse delivery. PI is a nucleic acid stain; when it enters the cell, it binds to RNA and DNA, which increases its fluorescence intensity. An insert with cells, prepared according to Section 2.2.1, was first placed into an empty well of a 24-well plate, where the growth medium was aspirated from the insert, the cells were washed with 300 μL of physiological saline, 300 μL of LCIS was added to the insert, and the insert was transferred to a well containing 500 μL PI staining solution and the bottom electrode. The top electrode was then immersed in the solution within the insert. The bottom electrode was positive to enable electrophoretic transfer of the positively charged propidium from the lower solution through the substrate pores into the cells. Prior to pulse delivery, the 24-well plate with the insert was placed on the microscope stage. Time course PI fluorescence increase before and after pulse delivery was monitored at 10× objective magnification with an inverted epifluorescence microscope Leica DMi8 Thunder Imager using the Leica LAS-X software environment. The PI dye was excited with Leica LED8 illumination source at a wavelength of 555 nm and its emission was passed through the DFT51010 filter cube and an additional 642/80 nm bandpass filter and detected with sCMOS camera Leica DFC9000 GT (all from Leica Microsystems, Wetzlar, Germany). The change in fluorescence intensity was determined in ImageJ Fiji [31] by averaging the pixel intensity in equally sized regions of interests (see Appendix A for further details) after processing the image with Leica’s instant computational clearing [32] in LAS-X software.

#### 2.2.6. Transfection

Inserts with cells, prepared according to Section 2.2.1, were transferred to a well prefilled with plasmid solution (1 mL of 100 µg/mL solution or 0.4 mL of 500 µg/mL solution), into which the bottom electrode was immersed. A lower volume was used in the case of higher plasmid concentration due to limited availability of the plasmid. The top electrode was immersed into the insert containing the growth medium in which the cells were grown (no washing steps were added to minimize the protocol). To enable electrophoretic transfer of the negatively charged plasmid into the cells, the top electrode was positive. After the applied pulse, the insert was left in the plasmid solution for 1 min and then transferred to a new well with 800 µL fresh cell growth medium preprepared in a 24-well companion plate (Falcon #353504). The inserts were placed on top of flanges to allow for better exchange between the old and fresh growth medium on top and bottom side of the substrate, respectively. For each type of substrate (0.4 µm and 1.0 µm diameter pores) and for each set of experiments performed on a given day, one insert represented a sham control where no pulse was delivered, and the insert was only immersed in the plasmid solution for 1 min. After the experiments were completed, the cells were placed in an incubator and after 24 h eGFP expression was visually inspected under an inverted epifluorescence microscope (described in Section 2.2.5) and quantified with flow cytometer (Section 2.2.7). For epifluorescence imaging, eGFP was excited at 475 nm and the emission was passed through filter cube DFT51010 and an additional 535/70 nm bandpass filter. To obtain images shown in Figure 5a, cells within an insert, ~24 h after gene transfection, were stained with Hoechst 33342 (10 mg/mL solution in water, #H3570, Invitrogen) by incubating them with 13.5 µL Hoechst in growth medium for 7 min at 37 °C, upon which the staining solution was replaced with 300 µL LCIS and the cells were imaged. Hoechst was excited at 390 nm and the emission was passed through filter cube DFT51010 and an additional 535/70 nm bandpass filter. ImageJ Fiji [31] was used for visualizing and preparing the images.

#### 2.2.7. Flow Cytometry

To prepare the cells in inserts for flow cytometry, the inserts were first positioned into an empty well, washed with 300 μL of physiological saline and trypsinized with 100 μL trypsin-EDTA for 5 min at 37 °C to completely detach cells from the substrate. After 5 min, 100 μL (0.4 μm substrate pores) or 130 μL (1.0 μm substrate pores) of growth medium was added to each insert to neutralize trypsin activity. Greater volume had to be used for substrates with 1.0 μm pores due to the considerable drainage of liquid across the substrate. The liquid in the inserts was pipetted to form a homogeneous cell suspension before transferring 180 µL to a 1.5 mL tube. The tubes were centrifuged at 4 °C and 200× *g* for 5 min. Carefully and slowly, 160 µL of the supernatant was removed from the tubes with a pipette, and the remaining cell pellet was resuspended in 200 μL of PBS for measurements on a flow cytometer (Attune NxT, Life Technologies, Carlsbad, CA, USA). For each sample, we recorded 8000−10,000 events. The excitation source was a 488 nm laser, and the emitted fluorescence was passed through a 530/30 nm bandpass filter. The gate for flow cytometry protocol was set based on two negative controls (cell culture inserts that were untreated and inserts that were only immersed in DNA solution). The gating strategy is shown in Appendix A. The viable cells within the gate were defined as eGFP positive. Obtained data were analyzed with the Attune NxT software. All experiments were repeated 3–4 times.

#### 2.2.8. Cell Viability

The cell viability was assessed with MTS assay (CellTiter 96^®^ AQueous One Solution Cell Proliferation Assay, Promega, Madison, WI, USA). The solution was prepared according to the manufacturer’s instructions. Cells were plated into inserts as described in Section 2.2.1, and gene electrotransfer was performed as described in Section 2.2.5. The cell viability was measured ~24 h after gene electrotransfer in the following steps. The inserts were first moved into an empty well of a 24-well plate. The cell growth medium within the inserts was replaced with 240 μL of cell growth medium containing 40 μL of MTS reagent. Cells were incubated for 4 h at 37 °C and 5% or 10% CO_2_ (see Section 2.2.1). After 4 h, we transferred 120 μL (0.4 μm pores) or 50 μL (1.0 μm pores) from each insert into a well of a 96-well plate. A lower volume needed to be used for substrates with 1.0 μm pores, since more than half of the volume within the insert leaked out through the pores during the incubation period. In comparison, the leakage through 0.4 μm pores was very small. The absorbance of the samples within 96-well plates was detected at 490 nm with a microplate reader (Infinite M200, Tecan, Männedorf, Switzerland). The percentage of viable cells was determined as the ratio between absorbance in experimental samples and sham control, after subtracting from all values the absorbance of blank wells containing growth medium without cells. We confirmed that the measured absorbance is linearly related to the number of cells within the inserts (Appendix A). All experiments were repeated three times. We note that, on the one hand, the metabolic MTS assay can yield somewhat better survival than viability assays based on counting live cells [33]. On the other hand, production of the transgene is energy consuming, which can somewhat underestimate the measured cell viability. Nevertheless, we chose MTS for our study to be able to directly compare the obtained cell viability results from gene electrotransfer using conventional bulk electroporation in the same cell lines [34,35].

### 2.3. Statistical Analysis

SigmaPlot 11.0 (Systat Software Inc., San Jose, CA, USA) was used to analyze statistically significant difference between experimental groups exposed to different pulsing protocols using a one-way ANOVA and Bonferroni’s *t*-test (with alpha value 0.05) after testing data for normality (Shapiro–Wilk test with *p* value to reject 0.05) and equal variance (with *p* value to reject 0.05). Statistical analysis was carried out for each cell line separately.

## 3. Results

### 3.1. Numerical Model of Cell Electroporation on Porous Membrane

To approach the design of gene electrotransfer using cell culture inserts, we first built a model of a cell growing on top of a porous substrate. We considered substrates containing pores with diameter of 0.4 µm, 1.0 µm, and 3.0 µm, as such membranes are embedded in cell culture inserts that are commercially available, see for example [30].

Numerical modeling confirmed that the electric field becomes localized within substrate pores. The electric field strength within the substrate pores is homogenous and approximately equal to the ratio between the voltage across the substrate, *U_subs_*, and the substrate thickness, *d_subs_* (Figure 2a). The electric field slightly extends out from the substrate pores, enough to reach the parts of the cell membrane which are in direct vicinity of the substrate pores. As an electric pulse is applied, the localized electric field induces electroporation, i.e., formation of pores/defects within the cell membrane (referred to as “electropores”). The time course of electropore formation, predicted by the model upon exposure to a 10-ms-long pulse, shows that the local density of electropores rapidly increases within the first microseconds and then remains roughly constant until the end of the pulse (Figure 2b).

Numerical results further show that the extent of cell membrane electroporation strongly depends on the substrate pore size and *U_subs_*, as can be seen in Figure 2c,d. Figure 2c shows the spatial distribution of the electropore density along the cell membrane, whereas Figure 2d shows the fraction of the cell membrane where the electropore density exceeds 10^13^ m^−2^ (this value has been used as an approximate measure for detectable electroporation in previous modeling studies [28,36]). Figure 2d, in addition, compares the results for cells of two different sizes, their dimensions differing by a factor of 2. For substrate pores with a diameter of 0.4 µm, electroporation onsets at *U_subs_* of about 5–7.5 V and remains localized to the vicinity of substrate pores up to ~15 V. For substrate pores with diameter of 1.0 µm, electroporation onsets at *U_subs_* of about 2.5–5 V, but already at 7.5 V, more than 15% of the cell membrane becomes electroporated. This means that the window of *U_subs_* enabling localized electroporation is considerably narrower for 1.0 µm compared to 0.4 µm substrate pores. With a further increase in *U_subs_*, regardless of the substrate pore size, almost the entire cell membrane surface becomes electroporated, including the surface of the cell that is not in contact with the substrate. For substrates pores with a diameter of 3.0 µm, electroporation also onsets at *U_subs_* of about 2.5–5 V, but already at 7.5 V, roughly half of the cell membrane becomes electroporated. Since electroporation of an excessively large cell membrane area can be detrimental to the cell, results in Figure 2d demonstrate two main findings. First, *U_subs_* needs to be carefully optimized for gene electrotransfer using cell culture inserts in order to achieve localized electroporation and prevent cell damage. Second, substrates with 3.0 µm pores are unsuitable for localized electroporation; thus we excluded them from further testing.

### 3.2. Electric Potential Distribution within the Full Insert System

Based on results presented in Section 3.1, we continued to numerically test electroporation in inserts with substrates containing pores of 0.4 µm and 1.0 µm. The substrate is only a part of the insert, meaning that *U_subs_* is generally not equal to the voltage *U_app_* applied to the electrodes, *U_subs_* ≠ *U_app_*. Indeed, our calculations, considering the full insert system, show that *U_subs_* presents a fraction of the applied voltage, which depends on the electrode geometry, especially that of the bottom electrode. Figure 3a shows the results for inserts with 0.4 µm substrate pores. When the bottom electrode is simply dipped at one side of the insert (Figure 3a,I), *U_subs_* is inhomogeneous and varies between ~20–40% of *U_app_*. A ring-shaped bottom electrode results in higher, but still highly inhomogeneous *U_subs_*, where the highest values of *U_subs_* are concentrated around the edge of the substrate (Figure 3a,II). The solution to a homogenous *U_subs_* is found with a spiral bottom electrode. Provided that the spiral has more than four turns (for the given well geometry), *U_subs_* is homogeneous along the entire substrate and is equal to ~50% of *U_app_* (Figure 3a,III,IV). The spiral electrode is equivalent to a plate electrode (Figure 3a,V). For spiral bottom electrode, the shape of the top electrode does not have a major influence on *U_subs_*. Similar results are obtained for inserts with 1.0 µm substrate pores, except that the *U_subs_* for spiral bottom electrode presents a smaller fraction of *U_app_*, ~20% (Figure 3b,IV). According to these results, we decided to carry out experiments using a spiral bottom electrode with seven turns and an L-shaped top electrode.

### 3.3. DNA Translocation Time

Before proceeding with experiments, we needed to choose an appropriate duration of the applied electric pulses, which we determined by considering the translocation time of plasmid DNA across substrate pores. Given the diameter of the substrate pores (0.4 µm and 1.0 µm), we assumed that the plasmid DNA can translocate without unfolding. This reasoning is based on the information that the radius of gyration of a circular plasmid DNA with ~5900 base pairs is *R_g_* = 0.131 µm (for YOYO-1 stained DNA molecule) and *R_g_* = 0.110 µm (for an unstained DNA molecule) [37]. Consequently, we can estimate the time needed for a DNA molecule to travel the substrate thickness as:(7)$$t_{trans}=\frac{d_{subs}}{\mu_{pDNA}E}=\frac{d_{subs}{}^{2}}{\mu_{pDNA}U_{subs}}$$
where *E* is the electric field within the substrate pores, which can be approximated as *U_subs_*/*d_subs_*, and *µ_pDNA_* is the electrophoretic DNA mobility. Under assumption that the DNA does not unfold, we can use the value for the bulk DNA electrophoretic mobility reported as (3.75 ± 0.04) × 10^−4^ cm^2^/(Vs) at 25 °C [38]. For *U_subs_* of 2.5–5 V, which is approximately the minimum required for achieving cell membrane electroporation (Figure 2d), the translocation time *t_trans_* is 1.1–0.5 ms. A recent statistical framework suggested that the minimal number of plasmid DNA molecules in the nucleus for transgene expression is in the order of 10 [39]. Therefore, we decided to set the pulse duration to 10 ms. Such pulse duration has often been used also for nanochannel-based gene electrotransfer [14,16,18].

### 3.4. Final Experimental Configuration and Model Validation

The final experimental insert system is shown in Figure 4a–c. To validate our numerical calculations, we first compared the computed and measured electrical resistance of the insert system. The computed resistance for substrates with 0.4 µm and 1.0 µm pores, using model configuration IV from Figure 3, is 159 Ω and 91 Ω, respectively. This compares well with the experimental values of 164 ± 28 Ω and 110 ± 18 Ω, respectively, which were determined as the voltage-to-current ratio measured during exposure to a 10 ms pulse. Note that these measurements correspond to inserts with cells. Appendix A shows that the presence of cells, at densities which we used for electroporation, does not influence the insert system resistance considerably.

Furthermore, our modeling predicted that when using substrates with 0.4 µm pores, electroporation should occur for *U_app_* greater than ~10 V in (corresponding to *U_subs_* of ~5 V). To confirm this experimentally, we monitored the intracellular delivery of PI, a small membrane-impermeable molecule which fluorescence increases upon binding to nucleic acids inside the cytoplasm and is commonly used for detecting electroporation [40]. Cells in inserts were placed on the microscope stage and were exposed to single or multiple 10-ms-long pulses. While electroporation was hardly detectable at *U_app_* = 10 V, the average cell fluorescence became considerably more profound at *U_app_* = 20 V (Figure 2d). The density of plated cells did not influence the pulse amplitude at which electroporation was detected (Appendix A). Application of multiple pulses resulted in proportionally greater increase in cell fluorescence. However, the PI uptake was not purely electrophoretic, since the intracellular fluorescence kept increasing for more than 2 min after pulse delivery.

### 3.5. Transfection with eGFP Plasmid

To test the insert system for its efficiency for gene electrotransfer, we placed 100 µg/mL solution of plasmid DNA coding for eGFP in the bottom well and used 10-ms-long pulses with different amplitude and number to deliver the DNA into CHO cells. Figure 5a shows a representative example of successfully transfected cells growing on the porous substrate. The percentage of transfected cells was determined 24 h after transfection with flow cytometry, whereas the cell survival was assessed with the metabolic MTS assay. In substrates with 0.4 µm pores (Figure 5b), successful transfection of ~7% of the cells was observed already when applying a single 10-ms-long pulse of 10 V. Using a single pulse, the highest transfection of ~12% was achieved at 20 V. These results are consistent with numerical modeling, which predicted that localized electroporation for substrates with 0.4 µm pores occurs for *U_subs_* between ~5–15 V, which is achieved at an applied voltage *U_app_* of ~10–30 V. Four 20 V pulses significantly improved transfection efficiency to ~30% compared to all single pulse exposures (*p* < 0.001); however, the transfection efficiency reached a plateau when further increasing the pulse number, with no significant difference between 4, 8, or 12 applied pulses. For all tested pulse parameters, the average cell viability was >90%; a trend towards lower viability with increasing pulse number could be observed. In substrates with 1.0 µm pores (Figure 5c), the transfection efficiency was on average comparable to substrates with 0.4 µm pores; however, it turned out to be highly variable with no significant difference between the experimental groups. High variability was also observed in the measured cell viability. We attribute this variability partially to the narrow window of *U_subs_* suitable for localized electroporation (Figure 2d) and greater influence of the random variability in electrode position compared to substrates with 0.4 µm pores (Appendix A). Furthermore, we noticed that the 1.0 µm substrate pores are large enough to result in considerable drainage of the liquid across the substrate on minutes time scale, mixing the top and bottom liquid, which unavoidably influenced the reproducibility of the experiments. Due to these challenges, we suspended substrates with 1.0 µm pores from further testing.

The transfection efficiency obtained in CHO cells grown on substrates with 0.4 µm pores was encouraging. Therefore, we tested the transfection efficiency in two other cell lines, rat cardiac myoblasts H9C2 and mouse myoblasts C2C12 (Figure 6). We first transfected the H9C2 and C2C12 cells with four 20 V pulses, which worked best for CHO cells; however, we observed that 20 V can already be quite damaging to H9C2 cells. This could be related to the larger size of these cells compared to CHO cells, since our numerical calculations suggest a tendency of larger cells to become more electroporated (Figure 2d). By reducing the applied voltage to 15 V and 10 V, the cell viability was preserved at or above 81% and 98% for H9C2 and C2C12 cells, respectively. Nevertheless, the average transfection efficiency remained below 10%.

To test whether an increase in plasmid concentration improves gene expression, we performed additional experiments raising the plasmid concentration from 100 µg/mL to 500 µg/mL [34,35]. Indeed, we obtained better transfection in all cell lines. The percentage of transfected cells reached on average 44%, 21%, and 30%, for CHO, H9C2, and C2C12 cells, respectively, when exposing them to four 10 ms pulses of 20 V (CHO) or 15 V (H9C2 and C2C12) (Figure 7). Increasing the number of pulses to eight resulted in similar transfection efficiency as with four pulses (24 ± 11% for H9C2; 26 ± 5% for C2C12).

## 4. Discussion

The concept of localized electroporation offers the possibility to transfect cells with high efficiency and high viability, which is indispensable to gene therapeutic applications. However, localized electroporation generally involves fabrication of nanoscale structures (pores/channels or straws) on substrates and requires sophisticated clean-room facilities that preclude its use in budget and resource limited laboratories or from laboratories that do not have access to such facilities. Recently, it was shown that localized electroporation can be achieved with low-cost commercially available polycarbonate porous substrates containing 0.1 µm or 0.2 µm diameter pores [21,22]. Nevertheless, such substrates require custom coating to enable cell attachment and custom device fabrication for electroporation, and sterilization. Furthermore, polycarbonate substrates are opaque, making them incompatible with inverted optical microscopes, which are often the only type of microscopes used in cell biology laboratories. In our study, we tested whether similar transfection efficiency and benefits of localized electroporation could be achieved with PET porous substrates embedded in cell culture inserts, which are also commercially available. The advantages of such inserts are that the substrate is already precoated for optimal cell attachment, PET substrates are transparent, enabling cell visualization with inverted microscopes in transmission and fluorescence mode, and the substrate requires no specific device assembly for electroporation—the porous substrate is pre-embedded at the bottom of a holder that can be inserted into a well of a multiwell plate and can be easily combined with wire electrodes for electroporation.

Our theoretical analysis showed that cell culture inserts with porous substrates, even though their minimum pore diameter is 0.4 µm, enable localized electroporation. Through experiments on CHO cells, we confirmed that we are able to transfect cells under conditions which were predicted by theoretical modeling to result in successful transfection. With minimal optimization, such as increasing the number of pulses from 1 to 4, we could increase the average transfection efficiency of CHO cells from around 10% to 30%. We could further increase the transfection efficiency to 44% by increasing the concentration of the plasmids from 100 µg/mL to 500 µg/mL. Similar optimization in C2C12 and H9C2 cells resulted in an increase in average transfection efficiency from below 10% to around 20–30%, at 100 µg/mL and 500 µg/mL plasmid concentration, respectively. Therefore, we are confident that with further optimizations of plasmid concentration, plasmid size, and possibly the solution in which the plasmid is dissolved, pulse parameters, temperature, etc., one can achieve even greater transfection efficiency.

The transfection efficiencies achieved in our study are already comparable to that achieved with conventional bulk electroporation in the same cell lines using the same plasmid and protocols of experimental analysis (flow cytometry and MTS assay). Specifically, Potočnik et al. [34] investigated the influence of pulse parameters and plasmid concentration on gene electrotransfer in CHO cells using conventional electroporation. When using 8x 5 ms pulses and plasmid concentration ranging between 100–500 µg/mL, similarly as in our study, they achieved an average transfection efficiency of 50–55%, while keeping the cell viability at 65–75%. In a subsequent paper studying C2C12 cells, the maximum transfection efficiency and cell viability varied, respectively, between 20–40% and 80–90% [35]. Note that the cell viability obtained is our study (99% for CHO and 98% for C2C12), for pulse parameters yielding the highest transfection efficacy, is considerably better compared to conventional electroporation. Improved viability is at least to some extent related to the fact that localized electroporation can be performed with low-voltage pulses. Lower voltages are associated with lower electric currents flowing through the sample, and consequently, reduced electrochemical reactions, which helps with retaining cell viability [41]. Another advantage of low-voltage pulses is linked to the complexity and cost of equipment needed to generate such pulses. Conventional electroporation requires the use of high-voltage pulse generators with complex power electronics design that cost around or above 10,000 EUR/USD (for academic use). On the contrary, the 10 ms, 15–20 V pulses used in our study can easily be generated using a simple electrical circuit driven by a low-voltage (battery) supply, as shown in Appendix A. The low cost, ease of access, and ease of assembly of such a pulse generator provides a benefit to laboratories that might not want to commit to an expensive high-voltage pulse generator, e.g., where the use of electroporation is infrequent.

However, the transfection efficiencies and cell viability (for H9C2 cells, ~81%) achieved in our study are lower compared to the system of Cao et al. [22], which enabled up to 50–80% transfection efficiency while preserving >95% cell viability. It should be noted that ours and Cao et al. study cannot be directly compared, since Cao et al. used different substrates with different thickness, pore size, porosity, and material (polycarbonate), as well as different electrode geometry, pulse parameters (trains of 200-µs-long pulses), and cell lines (HeLa and Jurkat). All these parameters can influence localized electroporation and gene electrotransfer efficacy to some extent, as shown through experiments and theoretical modeling in our and previous studies [14,21,22,42]. Our theoretical models allow us to discuss how the substrate pore size can have an important influence on plasmid delivery, expression and cell viability through four possible mechanisms, as follows.

Firstly, our calculations of the electroporated area fraction show that substrate pores with 0.1 µm diameter enable localized electroporation (small fraction of electroporated cell membrane) over a considerably wider range of *U_subs_*, compared to substrate pores with a diameter ≥0.2 µm (Figure 8a). The small range of voltages enabling localized electroporation in larger substrate pores means that small variations in *U_subs_*, for example due to variation in the electrode position, can lead to some of the cells becoming extensively electroporated, bringing the behavior close to bulk electroporation. Secondly, pores with 0.1 µm diameter have a considerably greater *U_subs_* threshold, at which electroporation occurs (~25 V) compared to substrate pores with diameter ≥0.2 µm (~10 V), see the inset in Figure 8a. Since the electric field within the substrate pores is proportional to *U_subs_*, greater *U_subs_* directly means greater electrophoretic force that drags the DNA from one side of the substrate, through the substrate pores, and into the cell. Smaller substrate pores are, therefore, indirectly associated with a greater number of transported DNA molecules per unit time, provided that the DNA is small enough to fit into a substrate pore without unfolding. Thirdly, the size of the substrate pores has a major influence on the resistance of the substrate (Figure 8b). Substrate pores with 0.1 µm diameter result in more than an order of magnitude greater resistance compared to pores with 0.4 µm diameter, considering equal substrate porosity. Any system similar to ours can be understood as a voltage divider consisting of a resistor representing the electrodes together with the electrolyte solutions and a resistor representing the substrate. The greater the substrate resistance compared to the surrounding solutions, the greater the voltage that establishes on the substrate. This means better control over *U_subs_*, and thus, better reproducibility. In our system, the resistance of the substrate is comparable (0.4 µm substrate pores) or even lower (1.0 µm substrate pores) than the surrounding solutions, for which moderate variations in electrode positions have a considerable influence on *U_subs_*. (Appendix A). Finally, the size of the substrate pores has a major influence on pore areal fraction (Figure 8c), which is directly proportional to the diffusive flux of solutes across the substrate. Electrochemical reactions at the interface between a metal electrode and electrolyte are an unwanted, but hardly avoidable companion of electroporation. Indeed, when applying ~60 V to the electrodes in our system, we would often observe bubble formation. Greater pores in the substrate can, therefore, be associated with greater diffusive flux of damaging free radicals. By imaging the kinetics of electroporative PI uptake, we have, indeed, observed that PI continues to enter the cells beyond 2 min after pulse application (Figure 4d). Such a long-term increase in cell membrane permeability has been proposed to be associated with oxidative lipid damage caused by free radicals [43,44]. In contrast, PI delivery in classical nanochannel electroporation using a single nanochannel with diameter of ~90 nm [14] showed an immediate increase in cell fluorescence which stabilized within 2.3 s, suggesting no long-term post-pulse diffusive uptake of PI.

The analysis above calls for a study that systematically compares the influence of the size of substrate pores and porosity on transfection efficiency and cell viability. Currently, any conclusion or interpretation is hindered, since previous studies on localized electroporation used different cell lines, porous substates with different properties (material, thickness, porosity, and pore size), different geometrical configurations of the electroporation device, and different pulse parameters [14,15,16,17,18,19,20,21,22,23].

## 5. Conclusions

Our study shows that cell culture inserts with porous substrates enable localized electroporation and a plasmid electrotransfer efficiency that is comparable to that achieved with conventional bulk electroporation. One of the benefits of gene electrotransfer using such porous substrates is that it does not require expensive high-voltage pulse generators; the low voltage-pulses required for efficient plasmid transfection can be generated by a simple low-cost (battery supplied) electrical circuit (Appendix A). The benefit of cell culture inserts over previous electroporation systems based on commercial porous substrates [21,22] is that they are already sterilized and precoated for optimal cell attachment, they are pre-attached to a holder, and consequently, require no additional (micro)fabrication procedures to assemble the insert for electroporation. Our system for gene electrotransfer simply consists of an insert, a multiwell plate, and a pair of platinum/iridium wires as electrodes. Furthermore, unlike polycarbonate substrates used in previous studies, we chose inserts with transparent PET substrates, which are compatible with inverted optical microscopes both in transmission and fluorescence mode and allow for easy cell visualization. However, the smallest size of substrate pores currently available with cell culture inserts is 0.4 µm, which according to our theoretical predictions might be suboptimal for gene electrotransfer. Further systematic studies are needed to study the influence of substrate pore size and porosity on the efficiency of gene electrotransfer and cell viability. We hope such further studies will inspire the production of cell culture inserts with substrates containing smaller pores for gene electrotransfer applications.

## Figures and Tables

**Figure 1 pharmaceutics-14-01959-f001:**
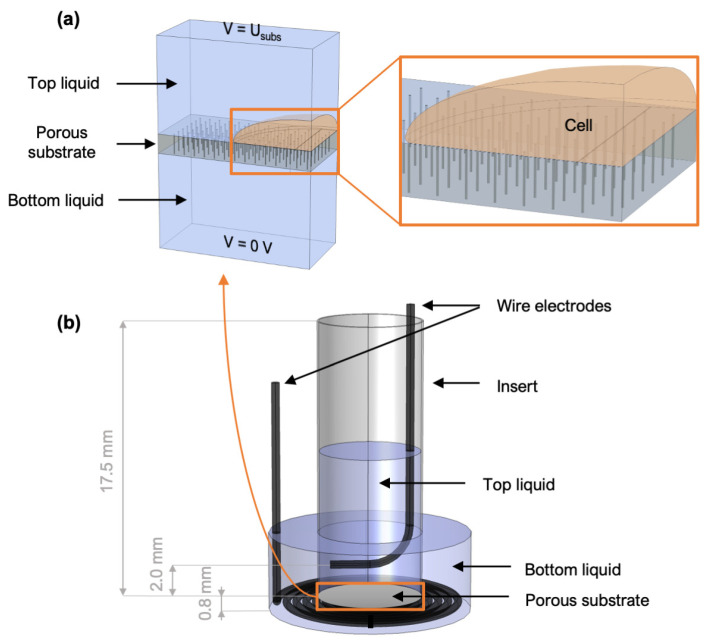
Geometry of the numerical models. (**a**) Model of a cell on top of a porous substrate. (**b**) Model of the full insert system.

**Figure 2 pharmaceutics-14-01959-f002:**
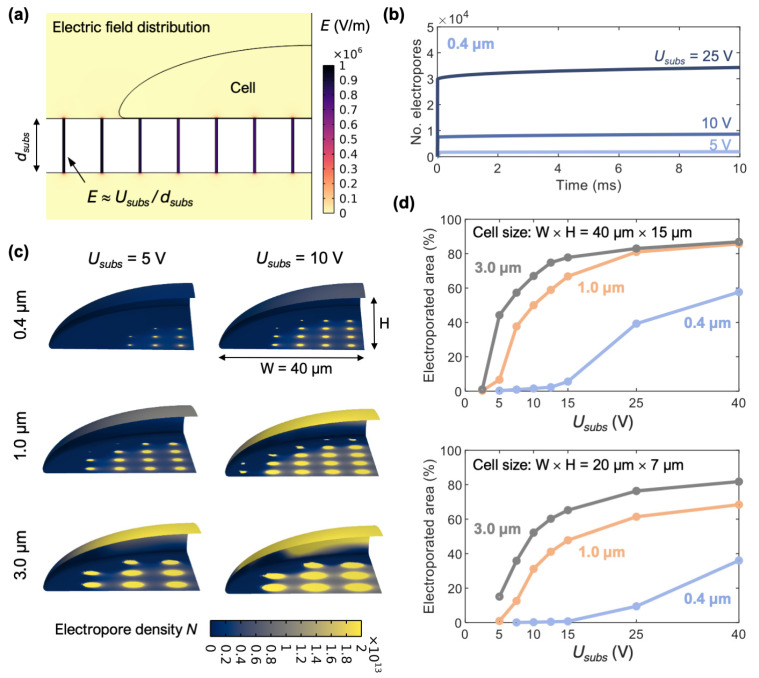
Numerical modeling investigating electroporation of a cell growing on porous substrate. (**a**) Electric field distribution within and around the substrate pores. (**b**) Time course of the total number of electropores induced by a 10-ms-long pulse of given *U_subs_* voltage applied at time 0 s. Results are shown for substrate with 0.4 µm pores and for cell with dimensions 40 µm × 15 µm. (**c**) Spatial distribution of the number density of electropores within the cell membrane at the end of a 10 ms pulse. Results are shown for substrates with three different pore sizes and for two different values of *U_subs_*. (**d**) Electroporated fraction of the cell membrane, determined as the relative area in which the electropore density exceeds 10^13^ m^−2^. The two graphs correspond to calculations for cells with two different sizes. The electroporated area was computed for *U_subs_* = {1, 2.5, 5, 7.5, …} V; absence of a data point at these values means that the electropore density is below 10^13^ m^−2^ everywhere in the cell membrane.

**Figure 3 pharmaceutics-14-01959-f003:**
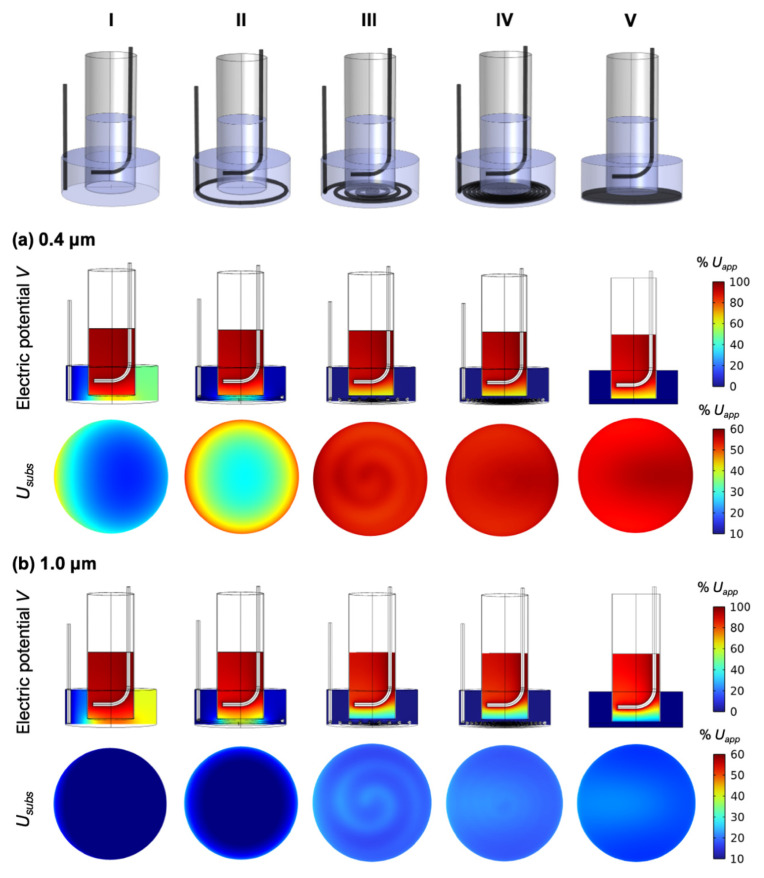
Influence of the electrode configuration. The images in the top row show the tested electrode configurations (**I**–**V**). (**a**) Electric potential distribution in the insert system and the voltage drop over the substrate *U_subs_* for the different electrode configurations. Calculations are for substrates with 0.4 µm pores. The electric potential and *U_subs_* are expressed as the fraction the voltage applied to the electrodes, *U_app_*. (**b**) Same calculations as in (**a**) but for substrates with 1.0 µm pores.

**Figure 4 pharmaceutics-14-01959-f004:**
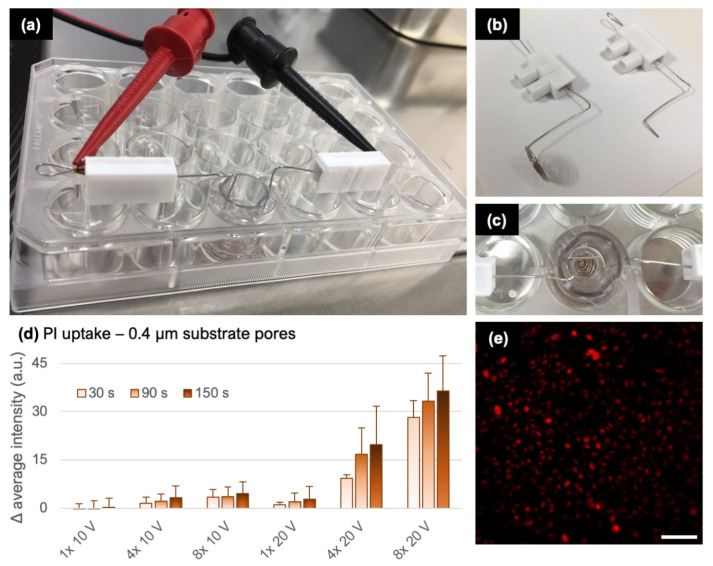
Final experimental configuration and testing. (**a**) Images of the 24-well plate containing an insert and wire electrodes. (**b**) Wire electrodes. (**c**) Insert positioned in the 24-well plate, viewed from top. (**d**) PI intracellular uptake detected by increase in the average sample fluorescence. Bars with different shades show the change in fluorescence at indicated time points after delivery of 10-ms-long pulses. The pulse number and amplitude are indicated at the *x*-axis. Graph shows the mean and range from two experiments. (**e**) Fluorescence image of cells stained with PI upon exposure to 4× 10 ms, 20 V pulses. Scalebar 100 µm.

**Figure 5 pharmaceutics-14-01959-f005:**
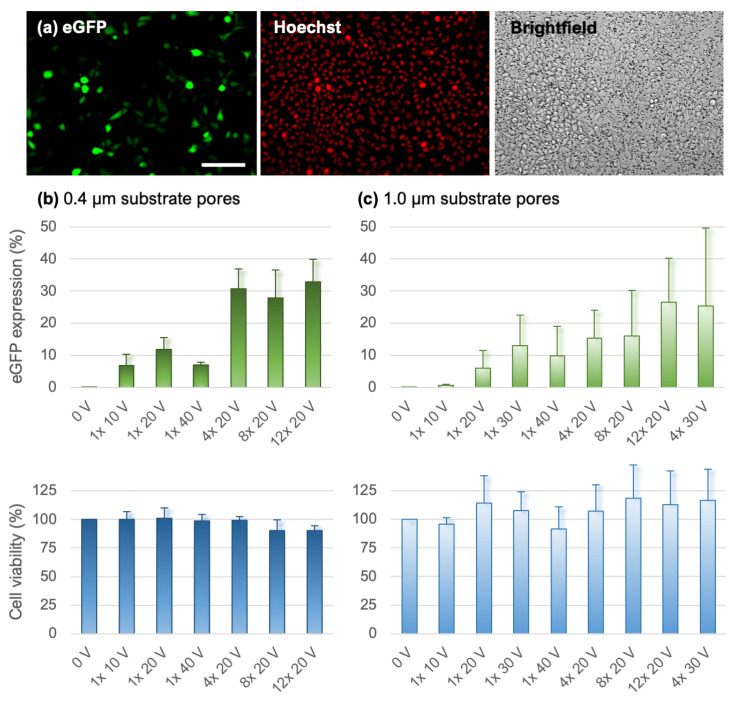
Transfection efficiency and cell survival in CHO cells. (**a**) Representative fluorescence image showing cells expressing eGFP (**left**), all cell nuclei stained with Hoechst 33342 (**middle**), and the corresponding brightfield image (**right**). Scalebar 100 µm. (**b**) Results obtained using substrates with 0.4 µm pores. Samples were exposed to 10 ms pulses of indicated number and amplitude. eGFP expression was determined as the percentage of fluorescent cells using flow cytometry. Cell viability was determined by the metabolic MTS assay. Graphs show mean ± s.d. from 3−4 experiments. (**c**) Results obtained using substrates with 1.0 µm pores in the same way as in (**b**).

**Figure 6 pharmaceutics-14-01959-f006:**
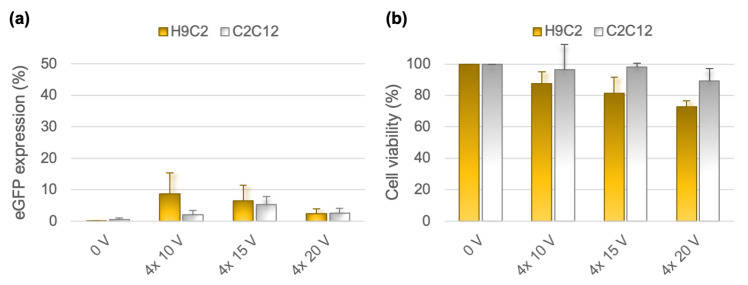
Transfection efficiency and cell survival in H9C2 and C2C12 cells. Experiments were performed in the same way as for CHO cells, but only using membranes with 0.4 µm substrate pores. (**a**) Percentage of cells expressing eGFP. (**b**) Percentage of viable cells. Graphs show mean ± s.d. from 3–4 experiments.

**Figure 7 pharmaceutics-14-01959-f007:**
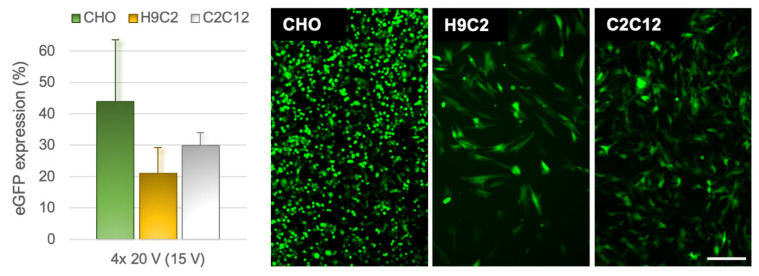
eGFP expression in CHO, H9C2, and C2C12 cell lines using higher plasmid concentration of 500 µg/mL. Transfection was performed with four 10-ms-long pulses with optimized voltages of 20 V (CHO) and 15 V (H9C2 and C2C12). Graph shows mean ± s.d. from 3–4 experiments. The fluorescence images visualize the corresponding eGFP expression in each cell line. Scalebar: 200 µm, same for all three images.

**Figure 8 pharmaceutics-14-01959-f008:**
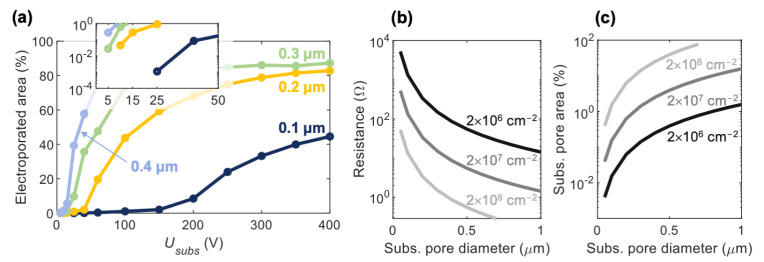
Calculations for substrate pores smaller or equal to 0.4 µm. (**a**) Electroporated fraction of the cell membrane, determined in the same was as in Figure 2d, but for substrate pores with diameters 0.1 µm, 0.2 µm, 0.3 µm, and 0.4 µm (for comparison). Porosity is 2 × 10^6^ cm^−2^. The inset shows at which *U_subs_* electroporation onsets for each substrate, i.e., at around 25 V, 10 V, 5 V, and 5 V for 0.1 µm, 0.2 µm, 0.3 µm, and 0.4 µm substrate pores, respectively. (**b**) Electrical resistance of substrates with different pore diameters. Results are shown for three different porosities, which cover the range of commercial porous substrates that have been used so far for localized electroporation [21,22]. (**c**) The areal fraction of substrate pores relative to the entire substrate area, depending on pore size and porosity. This areal fraction is directly proportional to the flux of solutes across the substrate.

**Table 1 pharmaceutics-14-01959-t001:** Values of model parameters.

Parameter	Symbol	Value	Reference
Cell height and largest semiaxis	-	15 µm, 40 µm 7 µm, 20 µm	Arbitrary
Extracellular liquid conductivity	*σ_e_*	1.5 S/m	Measured ^1^
Intracellular liquid conductivity	*σ_i_*	0.5 S/m	[29]
Cell membrane conductance	*G_cm_*	2 S/m^2^	[29]
Cell membrane capacitance	*C_cm_*	0.01 F/m^2^	[29]
Cell membrane thickness	*d_cm_*	5 nm	[29]
Electropore radius	*r_p_*	1 nm	[29]
Electropore conductivity	*σ_p_*	(*σ_e_*-*σ_i_*)/ln(*σ_e_*/*σ**_i_*)	[27]
Electroporation constant	*q*	2.46	[29]
Electroporation parameter	*a*	10^9^ m^−2^ s^−1^	[29]
Characteristic voltage of electroporation	*V_ep_*	0.25 V	[29]
Equilibrium pore density	*N* _0_	1.5 × 10^9^ m^−2^	[29]
Substrate thickness	*d_subs_*	10 µm	[30]
Porosity of substrate with 0.4 mm pores	*ρ_subs_*	2.0 × 10^6^ cm^−2^	[30]
Porosity of substrate with 1.0 mm pores	*ρ_subs_*	1.6 × 10^6^ cm^−2^	[30]
Porosity of substrate with 3 mm pores	*ρ_subs_*	0.8 × 10^6^ cm^−2^	[30]
Eff. conductance of substrate with 0.4 mm pores	*G_subs,eff_*	366 S/m^2^	Equation (6)
Eff. conductance of substrate with 1.0 mm pores	*G_subs,eff_*	1748 S/m^2^	Equation (6)

^1^ Conductivity of Live Cell Imaging Solution measured with conductometer SevenCompact (Metler Toledo).

## Data Availability

The models used for this study are available at https://github.com/learems/EP-CellCultureInserts, accessed on 19 June 2022. Additional experimental data are available on request from the corresponding author.

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
