# Peer review of "Gene Electrotransfer into Mammalian Cells Using Commercial Cell Culture Inserts with Porous Substrate"

_pharmaceutics, 2022, doi:10.3390/pharmaceutics14091959_

Round 1
Reviewer 1 Report
This work is a good contribution to expanding the performing electroporation.
I have some comments.
Introductions: It is somewhat confusing, and includes some of the discussion.
There are also more works where the action of the electric field is increased with micropore, which is good to include. The very long sentences make reading more difficult.
Materials, Result and conclusion is ok.
I would replace the low-cost equipment by expand the frontier of electroporation.
Reviewer 2 Report
In this paper, Vindiš and co-authors present a novel plasmid DNA electrotransfer approach into mammalian cells based on commercially available cell culture inserts with polyethylene-terephthalate porous substrate. The objectives of the study are well defined, and the introduction provides a state of art with appropriate references. Overall I consider that this article contains enough robust data to evidence the conclusions.
Some Comments:
-The authors state that one of the factors that prevents the direct comparation of viability and transfection results with Cao's work was the fact that they used different cell lines (HeLa and Jurkat). What was the rationale for choosing other cell lines, since your study was based on Cao's work?
- Please correct the figure 4 d) because in the graph it is repeated 4x10ms 20V
Reviewer 3 Report
The manuscript is nicely put. I congratulate the authors on that. Modeling is performed nicely. Few cell lines are used.
However, I have only a few concerns:
Viability measurement methods. MTS is a metabolic activity this is why the authors might have misinterpreted the cellular viability since the production of the transgene is energy consuming. I presume authors should obtain even better viability once the colony counting test is done or even the cell count with the flow cytometer.
Another concern is the main message that is “we did it at low cost”. Such conclusions usually go in low-impact factor journals. I would focus more on lower current (means less viability change), miniaturization, etc.
There are a few minor style errors. In line 201 equation probably is not 2 but 5.
Why were cells grow for two days? Usually, it’s 24 hours. Especially with confluency like it is shown in fig 5 A brightfield.
It was said that plasmid solutions (probably stock solutions were 100 µg/ml and 500 µg/ml). How then it was diluted with cells, but the concentration stayed the same? There is probably a clarification needed. The same goes for the distance between electrodes.
Why the incubation was for 1 minute after electroporation? This may result in 10 to 20 % of viability loss.
A gating strategy must be provided since it is crucial for the obtained results.
Some figures must be redone: Fig 5, Fig 6, Fig 7 (please put the percentages on the name on Y axis but not on the Y axis numbers)
Where are the whiskers (Statistics) in Fig. 4 D?
Why Fig. 4 E is black and white filter and others are with colors? Why?
The Hoechst 33342 dye is not mentioned in Methods. How were the cells stained?
What is the magnification of fig 7 fluorescent images? Is the magnification on those pictures the same? There is a white line on C2C12 but in the captions it says 200 mm. Is that correct?
Please use the same fonts in fig 8.
I recommend making the same color on viability measurements and the same color on transfection efficiency rather than marking different colors on pore substrates. The presented way is confusing.
Was ImageJ used for imaging? If so authors should include that into the methods.
Round 2
Reviewer 3 Report
Congratulations. This manuscript is ready for publishing. No additional questions.